# A Review of Asbestos Bioweathering by Siderophore-Producing *Pseudomonas*: A Potential Strategy of Bioremediation

**DOI:** 10.3390/microorganisms8121870

**Published:** 2020-11-26

**Authors:** Sébastien R. David, Valérie A. Geoffroy

**Affiliations:** 1SOMEZ, Parc Marcel Dassault, 34430 Saint Jean de Vedas, France; sebastien.david08@gmail.com; 2Department of Biotechnologie et Signalisation Cellulaire, Université de Strasbourg, CNRS-UMR7242, BSC, ESBS, Illkirch, 67413 Strasbourg, France

**Keywords:** asbestos, minerals, siderophores, iron, *Pseudomonas*, weathering

## Abstract

Asbestos, silicate minerals present in soil and used for building constructions for many years, are highly toxic due primarily to the presence of high concentrations of the transition metal iron. Microbial weathering of asbestos occurs through various alteration mechanisms. Siderophores, complex agents specialized in metal chelation, are common mechanisms described in mineral alteration. Solubilized metals from the fiber can serve as micronutrients for telluric microorganisms. The review focuses on the bioweathering of asbestos fibers, found in soil or manufactured by humans with gypsum (asbestos flocking) or cement, by siderophore-producing *Pseudomonas*. A better understanding of the interactions between asbestos and bacteria will give a perspective of a detoxification process inhibiting asbestos toxicity.

## 1. Introduction

Asbestos is an industrial term referring to six naturally occurring fibrous silicate minerals from the serpentine and amphibole groups. Chrysotile represents the single asbestiform mineral species from the serpentine group while the amphibole group contains five varieties: crocidolite, amosite, tremolite, anthophyllite and actinolite. Due to their insulating, chemical and mechanical properties, asbestos was intensively used in many commercial products for over 30 years, with chrysotile representing 95 % of the world production [1]. However, because of its toxic effects in humans, asbestos was banned in many countries since the beginning of the 1980s. Indeed, exposure to asbestos fibers by inhalation can cause serious pathologies such as fibrogenesis of the lung, pleural calcification, mesothelioma and ovarian or digestive system cancers [2,3]. The oxidative stress induced by free radicals’ production due to the presence of iron in fibers, of up to 30 wt%, is typically correlated with asbestos toxicity [4,5]. Iron generates free radicals and reactive oxygen species (ROS) via the Fenton reaction causing DNA damage [6,7]. Today, renovation or demolition of buildings generates tons of asbestos waste that needs to be managed and disposed of appropriately according to regulations. Asbestos containing waste (ACW) is generally bagged and deposited in a controlled landfill, while the toxicity or potential health and environmental risk of asbestos fibers remain [8,9]. During the last two decades, studies described various biological interactions with raw asbestos and few focused on ACW. A better knowledge of biological asbestos fibers dissolution may contribute to the development of the eco-friendly management of asbestos waste to reduce asbestos-related environmental and health problems.

Various microbial processes contribute to the dissolution of minerals, which may represent a micronutrient source essential in many enzymatic processes for microorganisms [10]. Two main mechanisms are involved in microbial bioweathering of rock and mineral i) biophysical mechanisms such as penetration of filamentous microorganisms or biofilms; ii) biochemical mechanisms including a wide diversity of metabolite production and redox reactions [11,12]. Among the metabolites excreted, organic and inorganic acids, production of metal-complexing exopolysaccharides or biosurfactants, and siderophores are involved in mineral dissolution. Moreover, biophysical and biochemical mechanisms can act synergistically to influence biological-mineral alteration [12]. Fibrous silicate minerals, like asbestos, are no exception to bioweathering processes. Indeed, organisms have been isolated from various serpentine sites. Bacteria isolated from asbestos rocks or soil from several Indian mines decrease the iron content of asbestos [13]. Rhizosphere bacteria contribute also to the weathering process in serpentine soil [14]. Telluric bacteria such as *Bacillus mucilaginosus* induced a mineral dissolution with an interesting loss of crystallinity in the serpentine fibers along with a pH decrease, organic acids and ligand secretion [15]. Concerning fungi, *Verticillium* sp., *Paecylomyces* sp. and *Fusarium oxysporum*, isolated from chrysotile bearing rocks, were all able to release iron from asbestos fibers [16]. Moreover, *Verticillium* sp. presented a higher efficiency of magnesium and silicon bioweathering towards chrysotile fibers extracted from Italian mine, while others are less active in structural ion removal such as *Fusarium oxysporum* [17]. Most of the studied microorganisms are able to produce siderophores, which might be a common mechanism in fungal and bacterial weathering of native asbestos, leading to iron dissolution from fibers. Direct evidence of iron removal from raw chrysotile [18] or amphibole [19] fibers by siderophore has been shown. Interestingly, reduction in asbestos toxicity due to iron dissolution was evidenced for the commonly used varieties of asbestos chrysotile [18,20], amosite [21], and crocidolite [20,21,22]. Considering the tons of ACW generated, only a few studies focused on ACW bioweathering. For cement wastes, nitrifying bacteria causes the chemical degradation of asbestos-cement of an agricultural building by the production of nitric and nitrous acid, while biofilm formation, together with acid producing bacteria, contribute to a wall thickness decrease of the asbestos cement pipes [23,24].

Many studies investigated the role of the *Pseudomonas* genus in mineral interaction since these bacteria are widely present in soils and well-known to produce various siderophores in large amounts. Recent investigations revealed the involvement of their high-affinity iron acquisition systems in asbestos bioweathering processes. Before understanding the interactions between the *Pseudomonas* species and asbestos, few data were available, whereas many papers related the implication of this genus in mineral weathering such as clay or iron oxydes, for example. Furthermore, Pseudomonas is a genus that is already used in bioremediation processes of xenobiotics compounds (aromatic compounds, alkane, etc.) and is therefore a potential candidate for a biotechnology development [25]. This review is focused on chrysotile fibers bioweathering by siderophore-producing *Pseudomonas* encountered in most isolated ACW or raw asbestos.

## 2. Structure and Properties of Chrysotile Fibers

Asbestiform minerals are found in the bedrock at various locations around the world. Serpentinized ultramafic rocks and serpentinized dolomitic marbles are the main bedrocks of which chrysotile fibers are found [26,27]. In addition to bedrock composition, metasomatism, a process of altering the composition of a rock, either by the addition or subtraction of chemical elements, is necessary for asbestos formation. Indeed, this phenomenon is caused by an influx of silica-rich fluids into the rock under particular conditions of temperature and pressure [27]. Among the serpentine group of minerals containing in particular the antigorite and the lizardite, chrysotile represents a small percentage of the minerals encountered. As a consequence, chrysotile fibers are found as veins in serpentines [26]. 

Asbestos fibers are silicate minerals composed predominantly of silicon (Si) and oxygen (O) organized in silicate tetrahedra (SiO_4_), which may occur as double chains as in the amphibole, or in sheets as in chrysotile [26,28]. Chrysotile is a hydrated magnesium silicate with the approximate composition Mg_3_Si_2_O_5_(OH)_4_. The chemical composition of chrysotile varies according to mineral deposit. Indeed, substitution of magnesium and silicon can occur in chrysotile fibers. In the brucite layer, magnesium can be replaced by Fe^2+^, Mn^2+^ or Ni^2+^, while in the silicate layer, silicon may be substituted by Al^3+^ or rarely Fe^3+^ [26] (Figure 1). The crystal structure of chrysotile fibers consists of layers rolled in spiral form, composed of inner tetrahedral silicate layers and outer magnesium hydroxide octahedral layers (brucite) [26,29]. The silicate and brucite layers share oxygen atoms, where two out of every three hydroxyls from magnesium hydroxide octahedra are replaced by apical oxygens of the silica tetrahedral. Mismatch of O–O distances induces a curvature of the layers and the formation of a hollow cylinder having an average diameter of approximately 25 nm and composed of approximately 12–20 layers [26,29]. These cylinders, named fibrils, bunch together to form a chrysotile fiber.

Asbestos fiber minerals have been extensively used in many commercial products due to their unique physicochemical properties. Indeed, excellent physical properties such as the great tensile strength, resistance to heat and corrosion, poor heat, acoustic and electric conduction, generated a wide utilization as an insulation material. Besides those interesting properties, the chrysotile fibers are sensitive to acid pH compared to the amphibole fibers, due to the differences in the chemical composition and the structure. Indeed, the outer brucite layer is dissolved in an acidic medium, releasing magnesium and leaving a silica residue [9,26]. 

When inhaled chrysotile fibers are present in the lung and pleura, the lung clearance of the fibers is more rapid than amphibole due to longitudinal cleavage into fibrils, which can break and be phagocytized easier by the macrophages [30]. The chrysotile fibers’ half-life in the lungs can be measured in terms of month while it is in years for amphibole [3,31].

## 3. Asbestos Treatment Technologies

Nowadays, ACW are generally disposed of in landfill sites, which is a cheap waste management. This practice does not eliminate the toxicity of asbestos and its potential release. Thereby, various treatments are currently being developed or industrialized.

The asbestos solidification and stabilization were designed to decrease exposure to fibers. Indeed, the solidification immobilizes asbestos into inert material such as a cement matrix [32], resulting in a reduction of fiber release, the decrease of the exposed surface area and the reduction of both porosity and permeability of ACW. The asbestos stabilization is a process leading to the reduction of the fibers mobility through the addition of adjuvant [8,9]. The solidification and stabilization treatments can be used as pretreatment before landfill disposal. These processes are considered as a safe waste management strategy reducing fiber inhalation. However, they do not eliminate asbestos toxicity and do not result in a re-usable end-product. Moreover, a consequence may be an increase by 30–200% of the waste volume stored in landfills [9,33].

Given that asbestos fibers are unstable at high temperature, various thermal treatments were proposed. Indeed, chrysotile starts losing the hydroxyl groups at 500–600 °C and is transformed into forsterite, which recrystallizes at 820 °C [9]. This thermal treatment varies between raw asbestos and waste, with a higher temperature required for ACW decomposition, reaching 1200 °C [9,34]. Thus, the vitrification, as an example, destroys the asbestos fiber structure with the conversion of the waste in a stable and homogeneous glass without toxicity. However, extreme temperature is required (1200–1600 °C) resulting in an expensive and energy-intensive process [9]. On the other hand, alternative thermal treatments, such as hydrothermal or microwave, were proposed in order to decrease the temperature and time required. These processes, cheaper than vitrification, require however high energy and are responsible for toxic gas release during these treatments [8,9].

The amorphization of chrysotile can be obtained with mechanical treatment related to the energetic fragmentation caused by the milling. This energy destroys molecular bonds and disrupts the crystal structure of asbestos. The main advantage of this process is that the end-product can be reused for the preparation of mortars, improving mechanical properties of this product [8,9,35]. However, this treatment is more expensive than the thermal treatments.

As described in Section 2, the chrysotile fibers are sensitive to acid pH. As a consequence, many chrysotile waste treatments by acid attack have been proposed. As an example, the use of strong acids such as hydrochloric acid [36,37], sulfuric acid [38] or nitric acid [39] were used to treat waste. Other studies propose the use of weaker acids to overcome the problems of acid management, with the use of organic acids such as oxalic [40,41], acetic or formic acid [9].

## 4. Siderophore-Producing *Pseudomonas*

Iron (Fe), the fourth most abundant metal in the earth’s crust, is a transition metal essential for the growth of almost all living microorganisms [42]. Indeed, Fe has key functions in many biological processes such as electron transfer, oxygen metabolism or DNA and RNA synthesis [43,44]. However, in aerobic circumneutral environments, Fe is poorly bioavailable due to its limited solubility and the slow dissolution kinetics of iron-bearing mineral phases [42,45]. Consequently, many specific uptake strategies have been developed by microorganisms in particular the production of siderophores, small molecules (200–2000 Da) with a high affinity for Fe^3+^ produced in iron limited conditions [46,47,48]. To date, more than 500 different types of siderophores with different chemical structures are known [49] and present a very high affinity for iron, on the order of 10^23^ to 10^52^ M^−1^. In soil environments, the concentrations of siderophores range quite broadly from tens of micromoles to a few millimoles per liter [50]. 

Siderophores produced by soil microorganisms play significant roles in weathering soil minerals and biogeochemical cycling of Fe [46,48]. Among these microorganisms, siderophore-producing *Pseudomonas* are widespread bacteria in soil and known for mineral-weathering capacity. Indeed, Pseudomonads are ubiquitous Gram-negative bacteria known for their adaptability and metabolic diversity and consequently are able to colonize a wide range of niches [51]. Therefore, many members of *Pseudomonas* are soil bacteria, but some are plant pathogens or human pathogens such as *Pseudomonas aeruginosa* that causes nosocomial infections, other are Plant Growth Promoting Rhizobacteria (PGPR) and, therefore, are beneficial to their host-plants [52,53]. The genus *Pseudomonas* produces a wide variety of siderophores and most of them are detectable under iron starvation [54]. The best known siderophores produced by fluorescent pseudomonads are the fluorescent high-affinity peptide pyoverdines (PVD) [55]. Some *Pseudomonas* are also able to produce diverse secondary siderophores of lower affinity such as pyochelin (PCH) in the case of *P. aeruginosa* [56,57].

PVDs, yellow-green fluorescent pigments, are composed of three distinct structural parts: (i) a dihydroxyquinolone chromophore, which confers the yellow-green color and fluorescence to the molecule, (ii) a strain-specific peptide chain, comprised of 6–12 amino acids bound to its carboxyl group, and (iii) an acyl side chain composed of a dicarboxylic acid residue, which can be either succinate, malate, or their amide forms, or alpha-ketogluratate or glutamate, depending on the producing strain and growth conditions [58,59,60,61,62]. Based on these structural differences, various PVDs are well-described, constituting a large family composed of more than 100 different PVDs depending on their peptide chain and radical R [63]. Figure 2 gives three examples of PVDs structure, two of them produced by *P. aeruginosa* and *P. syringae* sharing a ring structure in their peptide chain compared to *P. mandelii,* which has a linear one. Due to the complexity of the PVD structure elucidation, a rapid and efficient siderotyping method allowed pseudomonad characterization and identification based on PVDs differentiation. This useful strategy developed in the 1990s is founded on the different isoelectrofocusing profile related to the length, nature, and presence of cycle rings in the peptide chain in the PVD structure together with incubation of cells with a labeled iron-siderophore complex [64]. PVDs binds ferric iron with a high affinity (Ka = 10^32^ M^−1^) in a 1:1 (PVD:Fe^3+^) stoichiometry via a catechol group and two hydroxamate or hydroxy-carboxylate groups [59,65] (Figure 2A–C). Besides the high efficiency of complexing Fe^3+^, PVD has also been shown to complex with 16 different lower affinity metals (Ag^+^, Al^3+^, Cd^2+^, Co^2+^, Cr^2+^, Cu^2+^, Eu^3+^, Ga^3+^, Hg^2+^, Mn^2+^, Ni^2+^, Pb^2+^, Sn^2+^, Tb^3+^, Tl^+^ and Zn^2+^) and it was therefore suggested that this chelation could also be a means to protect the cells from some toxic metals since the uptake pathway selectivity was demonstrated [66,67,68].

As a second siderophore produced by *P. aeruginosa*, the pyochelin is a 2-(2-o-hydroxyphenyl-2-thiazolin-4-yl)-3-methylthiazo-lidine-4-carboxylic acid which chelates Fe^3+^ with an affinity of 10^28.8^ M^−2^ [69] in a 2:1 (PCH:Fe^3+^) stoichiometry [65]. A tetra-dentate chelator is provided by one molecule of PCH and a bi-dentate chelator by the second PCH to complete the hexacoordinate octahedral geometry necessary for Fe^3+^chelation [70] (Figure 2D). Aside from iron, it has been shown that PCH form stable complexes with other metal cations [66,68].

Due to the importance of the large variety and amount of siderophores in soil, the role of siderophore as a general mechanism in the dissolution of minerals such as clays, iron oxides or asbestos silicates has been well-studied for many microorganisms [46,71,72]. Therefore, the essential element such as iron could be available via Fe^3+^-siderophore complex formation at the mineral surface and transfer in the soil solution for uptake by microorganisms or plants.

## 5. Role of Pyoverdine and Pyochelin in Asbestos Weathering

Depending on asbestos species, various metals can be present in their structure or can enter the silicate minerals composition via substitution depending on metal-rich soil where they are formed. The presence of Fe and Mg are examples of elements often present in those minerals that are also important and highly required for most living organisms.

Some studies investigated the role of *Pseudomonas* bacteria for their ability to chelate mineral nutrients from soil minerals. *Pseudomonas mendocina* as a non-fluorescent species promoted the dissolution of iron-bearing mineral such as hematite, goethite and ferrihydrite [73] while *P. aeruginosa* demonstrated its ability to withdraw iron from silicate clay smectite clay through a siderophore-driven mechanism [71] or vitrified bottom ash silicates [74]. *P. aeruginosa* is a ubiquitous fluorescent *Pseudomonas* found in various environments [51], the siderophore production and iron acquisition of which are well known [75]. Indeed, *Pseudomonas* share the same weathering process as many microorganisms that use siderophore production to overcome iron limitation. As presented in Section 3, *P. aeruginosa* produces two endogenous siderophores, a more energy demanding high affinity molecule, PVD [76] and a lower affinity siderophore, PCH [77].

Recently, studies of David et al. [72,78,79,80] corroborate that these siderophores are also involved in the dissolution of other silicate minerals such as asbestiform minerals. Indeed, asbestos is an iron source for various microorganisms. Some studies reported on the one hand, the ability of siderophore-producing organisms to weather asbestos, for example, fungi such as *Fusarium oxysporum* [81,82], *Verticillium leptobactrum*, and *Aspergillus fumigatus* [83], rhizospheric bacteria [14] or Gram-positive bacteria isolates from asbestos mine [13,84]. On the other hand, direct evidence of iron weathering from asbestos by siderophore was shown with deferoxamine [18,85], EDTA [78,85] or citrate [85]. However, it is only recently that the direct implication of siderophore in the dissolution of raw asbestos and ACW was demonstrated by the use of siderophore mutants and fluorescent protein labeling [78,79,80]. First, asbestos bioweathering by siderophore-producing *Pseudomonas* was demonstrated with direct evidence of iron removal by PVD (Figure 3). David et al. [78] clearly showed the key role of the siderophores PVD and PCH in raw asbestos weathering, in particular with the significant impact on iron dissolution of the absence of both siderophores (Figure 3).

The same results were successfully obtained for ACW, with evidence of iron removal from asbestos linked to a siderophore-driven mechanism varying according to the materials. Indeed, depending on the waste, the percentage of asbestos fibers can be very different, and fibers can be free or embedded in various matrices. Asbestos cement (AC), which represents 80% of the world production of asbestos [3,86,87,88] and which is found in pipelines, flat sheets, corrugated roof sheeting or insulation boards, contain a ratio corresponding to approximately 10% asbestos fibers embedded in 90% cement [88]. This material is therefore a compact compound, in contrast to other major waste products, asbestos flocking (AF) found in insulation in buildings, which contains 90% free asbestos fibers and 10% of gypsum [88]. In addition, the amount of iron varies between both ACW with higher iron content in AC than AF, linked to the supply of iron as contamination by the cement matrix. The PVD and PCH pathway is well known to be repressed by the presence of iron and this variation in the chemical composition resulted in a stronger repression of both siderophore pathways in the presence of AC compared to AF (Figure 4). Although variation was evident, both ACW repress the siderophore system, suggesting that dissolved iron was released from the waste.

As already demonstrated with raw asbestos, the siderophores PVD and PCH also play an important role in the iron removal from ACW. Indeed, the absence of both siderophores greatly reduces iron dissolution from AC and AF (Figure 5), but the involvement of each siderophore depends on waste. In the presence of AF, the absence of one of the two siderophores is compensated by the production of the other, while in the case of AC, the absence of one of the two siderophores affected iron removal, with a stronger effect in the absence of PCH (Figure 5). The various matrices and percentages of asbestos fibers can probably explain the siderophore-driven mechanism differences between ACW. Moreover, given the small size of PCH compare to PVD, the iron extraction site may also be different depending on siderophores.

The use of siderophores as chelating compounds can be considered as a major mechanism of asbestos bioweathering. However, given the various known mechanisms involved in microorganisms–minerals interactions and the complexity of the siderophore-containing culture supernatant, which may contain various metabolites, other mechanisms may be involved in asbestos alteration such as organic acids, biofilm or redox processes. Siderophores and organic acids have already been shown to function synergistically in mineral dissolution [48]. The influence of organic acids in chrysotile dissolution is also a hypothetical process, given that bacterial growth is not affected by the absence of PVD and PCH, concluding of a sufficient iron dissolution [79,80]. In addition, *Pseudomonas* are also well-recognized to colonize minerals via a biofilm formation. The impact of the biofilm is controversial in the literature with studies claiming an acceleration or an inhibition of the alteration [89]. For example, Aouad et al. [74] demonstrated that biofilm decreases the alteration rate of the glasses or bottom ashes. Therefore, the biofilm constitutes a complex environment, which has not yet been clearly investigated. 

## 6. Asbestos Waste Dissolution by Pyoverdines

As presented in Section 3, PVDs constitute a large family composed of more than 100 different PVDs depending on structural differences corresponding to variations in the length, nature and presence of cycle rings in the peptide chain, as well as the radical R [63]. In addition to their iron complexation capacity, PVDs are able to chelate other metals [66,68]. However, the specificity of each PVD towards iron and more generally to metal complexation is not yet known.

Recently, David et al. [72,80] investigated the efficiency of various PVDs to scavenge iron from ACW. According to materials, the more efficient PVDs to extract iron are different (Figure 6). Indeed, among 10 PVDs tested, PVD-containing supernatants from *P. mandelii* and *P. syringae* were more efficient for iron removal from AF and AC, respectively. Few data are available related to the affinity constant of PVDs, with the exception of the PVD produced by *P. aeruginosa*. However, given PVDs share the same ligand groups, the affinity constant might not be sufficiently different to explain the differences according to ACW. Indeed, David et al. [80] hypothesized that the various 3D structures between PVDs might be the influencing parameter of iron extraction ability. Thus, some structures could more easily access the structural iron present in the layers of chrysotile.

The long-term bioweathering of ACW by renewal cycles using apo-PVD solutions highlighted iron release over time for AF or AC fFiuwaste (Figure 7). A siderophore dissolution mechanism of ACW has been suggested [18,72]. During the first renewal cycles, PVDs probably adsorb to the fiber’s surface and might release iron present in the brucite layer due to magnesium substitution. Indeed, it is well known that siderophore-mediated iron removal involves siderophore adsorption on the iron-bearing mineral surfaces [46]. Exhaustion of surface sites could leach the fiber surface and potentially provide easier access to the iron present in the silica tetrahedron. Thus, the contact time and the PVD concentration play an important role in the iron dissolution mechanism of ACW, with an impact in siderophore surface coverage and the release of iron from the silica layer, which could be slower than in the brucite layer. As suggested by David et al. [78,80], the grinding may also promote accessibility of siderophores to iron on the surface, due to the increase of the specific surface area of asbestos.

Interestingly, David et al. [72,80] showed a large decrease of iron content in asbestos fibers after long-term alteration with PVDs and confirmed the active dissolution by the siderophores (Figure 8). This process of bioweathering could lead to a reduction in asbestos toxicity. Indeed, some studies established a relationship between the iron removal from asbestos fibers and the decrease of the amount of ROS generated by asbestos [16,90]. Therefore, further investigation could validate the use of siderophore-producing *Pseudomonas* as a potential strategy of bioremediation resulting in a lower fiber toxicity.

## 7. Conclusions and Prospects

Currently, asbestos removal is a worldwide concern and we have to deal with huge amounts of waste. Given the lack of satisfactory treatment, a better understanding of the interactions between asbestos and microorganisms provides interesting perspectives for a bioremediation strategy and the development of eco-friendly management of asbestos waste, avoiding the continuous disposal of ACW in landfills. The advantages of biotechnological methods are the low-cost and low energy demand of biological waste management.

Thus, the biodeterioration of ACW by siderophore-producing *Pseudomonas* appears to be a promising process. Indeed, *Pseudomonas* are able to use ACW as a nutrient source for its iron requirement via a siderophore-driven mechanism. Siderophores are clearly involved and play a key role in asbestos bioweathering processes. In addition, PVDs have the ability to progressively extract iron from both AC and AF waste with efficiency varying upon the PVD. Interestingly, a long-term treatment of ACW by PVDs leads to a significant reduction of iron content in asbestos fibers conducting probably to a lower fiber toxicity. Further investigation of the interaction between siderophore-producing *Pseudomonas* and asbestos could contribute to the development of a biotechnological process to treat asbestos waste and reduce asbestos-related environmental and health problems. 

## Figures and Tables

**Figure 1 microorganisms-08-01870-f001:**
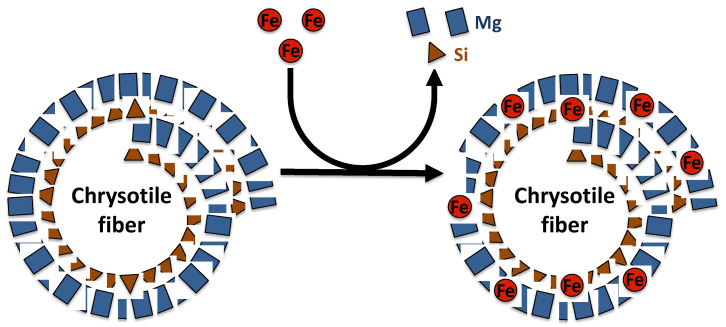
Representation of magnesium and silicon substitution by iron in the chrysotile fiber structure.

**Figure 2 microorganisms-08-01870-f002:**
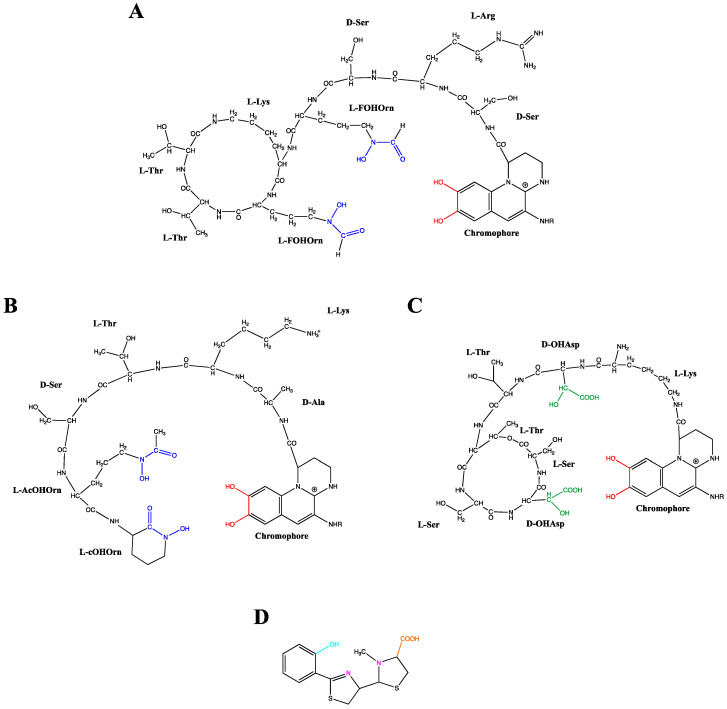
Structures of the pyoverdines from *Pseudomonas aeruginosa* (**A**), *Pseudomonas mandelii* (**B**) and *Pseudomonas syringae* (**C**) and structure of pyochelin from *Pseudomonas aeruginosa* (**D**). The pyoverdine ligand groups catecholate, hydroxamate and hydroxy-carboxylate are, respectively, highlighted in red, blue and green. The pyochelin ligand groups phenolate, imine/tertiary amine and carboxylate are respectively highlighted in turquoise, pink and orange. FOHOrn: δN-formyl-δN-hydroxy-ornithine; AcOHOrn: δN-acetyl-δN-hydroxy-ornithine; cOHOrn: cyclo-hydroxy-ornithine; OHAsp: threo-ß-hydroxy-aspartic acid.

**Figure 3 microorganisms-08-01870-f003:**
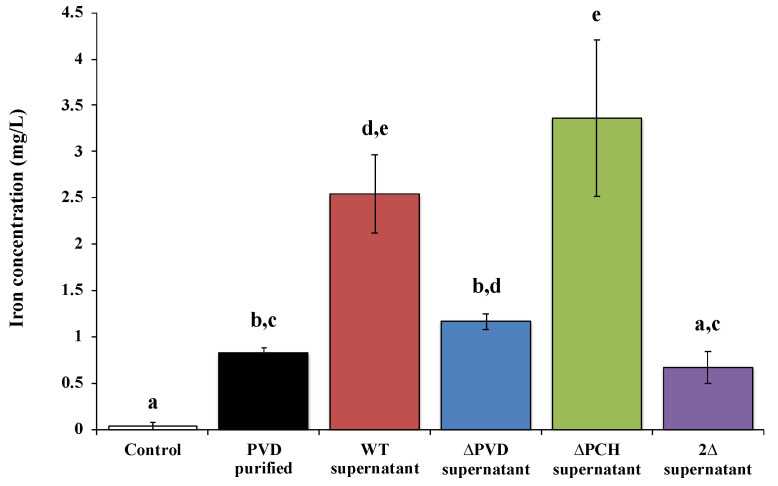
Concentration of iron dissolved from ground raw chrysotile (0.2 g) after 48 h incubation in the presence of succinate medium (control), 200 µM *Pseudomonas aeruginosa* PAO1 purified pyoverdine (PVD purified) or culture supernatant of the wild type *Pseudomonas aeruginosa* PAO1 strain producing both siderophores pyoverdine and pyochelin (WT supernatant), a pyoverdine-deficient strain (ΔPVD supernatant), a pyochelin-deficient strain (ΔPCH supernatant), or a pyoverdine- and pyochelin-deficient strain (2Δ supernatant). Error bars indicate the standard errors of the means of three replicates. Bars with the same letter are not significantly different (*p* > 0.05, Kruskal–Wallis test, three replicates). Adapted from David et al. [78].

**Figure 4 microorganisms-08-01870-f004:**
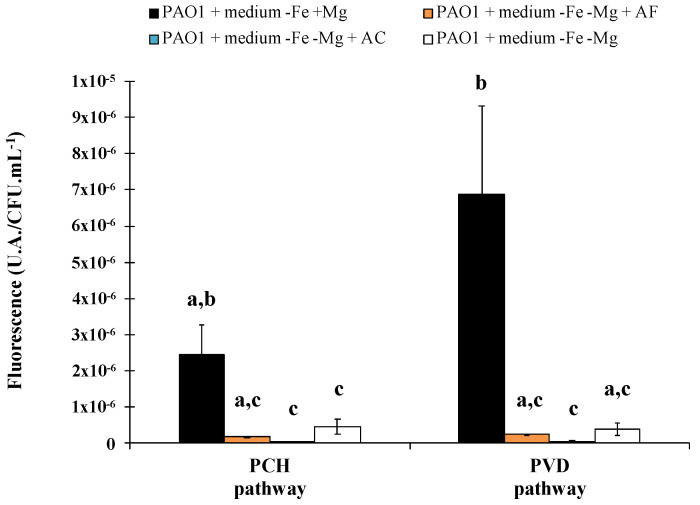
Biosynthetic enzymes for pyoverdine and pyochelin production were tagged with fluorescent protein mCherry to follow the synthesis of proteins involved in siderophore production (PVDJ for pyoverdine and PchA for pyochelin). Bars represent the expression of the pyoverdine and pyochelin biosynthetic pathways after 24 h of *Pseudomonas aeruginosa* PAO1 growth in casamino acids medium restricted in iron (−Fe) with (+Mg) or without (−Mg) magnesium, in the presence or absence of asbestos flocking (AF) or asbestos cement (AC). *P. aeruginosa* PAO1 growth in medium −Fe +Mg and −Fe −Mg corresponds, respectively, to the positive and the negative control. Error bars indicate the standard errors of the means of three, five or six replicates. Bars with the same letter are not significantly different (p ≥ 0.05, Kruskal–Wallis test, three, five or six replicates). Adapted from David et al. [79,80].

**Figure 5 microorganisms-08-01870-f005:**
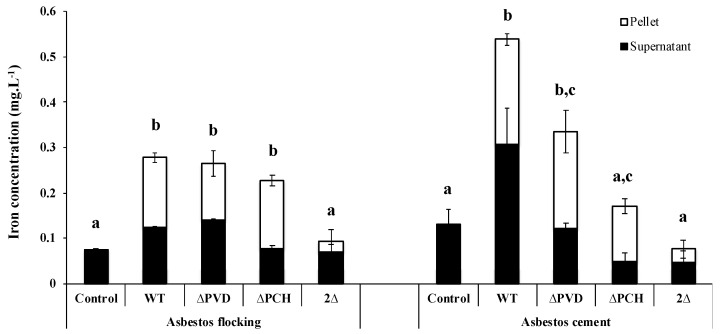
Concentration of iron dissolved after 18 h incubation from flocking asbestos waste or after 40 h incubation from asbestos cement in the presence of minimal casamino acids medium depleted in iron and magnesium as a control or inoculated with the wild type *Pseudomonas aeruginosa* PAO1 strain (WT), a pyoverdine-deficient strain (ΔPVD), a pyochelin-deficient strain (ΔPCH), or a pyoverdine- and pyochelin-deficient strain (2Δ). The iron was measured in the bacterial cells (pellet) and supernatants to determine the total amount of total iron extracted. Error bars indicate the standard errors of the means of three replicates. Bars with the same letter are not significantly different (*p* ≥ 0.05, one-way ANOVA for flocking asbestos waste and Kruskal–Wallis test for asbestos cement, three replicates). Adapted from David et al. [79,80].

**Figure 6 microorganisms-08-01870-f006:**
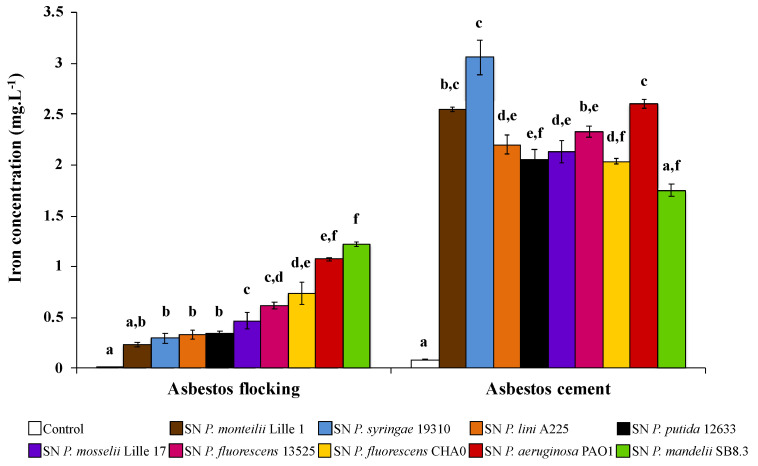
Concentration of iron dissolved from asbestos flocking or asbestos-cement in the presence of 100 μM of various pyoverdine-containing supernatants (SN) after 24 h of contact. Error bars indicate the standard errors of the means of three or five replicates. Bars with the same letter are not significantly different (*p* ≥ 0.05, Kruskal–Wallis test, three or five replicates). Adapted from David et al. [72,80].

**Figure 7 microorganisms-08-01870-f007:**
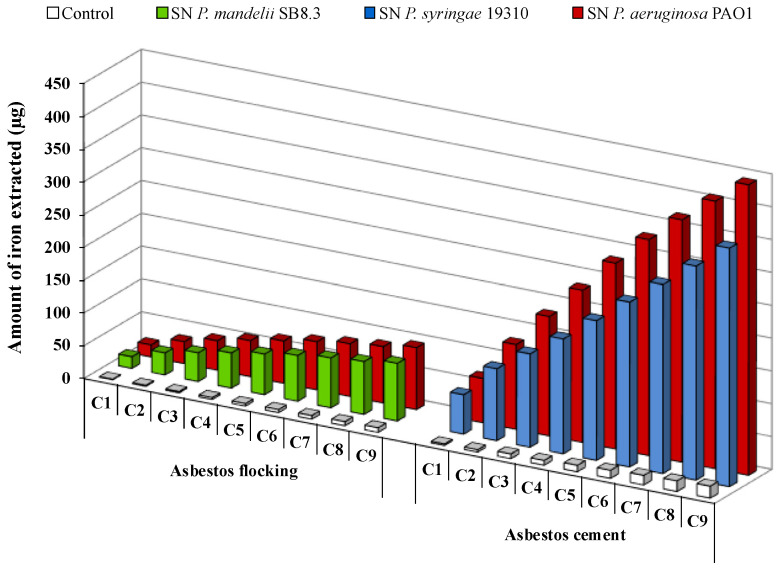
Iron removal from asbestos flocking or asbestos cement after 24 h renewal cycles (C1 to C9) in the presence of 100 μM of *Pseudomonas mandelii*, *Pseudomonas aeruginosa* or *Pseudomonas syringae* pyoverdine-containing supernatants (SN). Adapted from David et al. [72,80].

**Figure 8 microorganisms-08-01870-f008:**
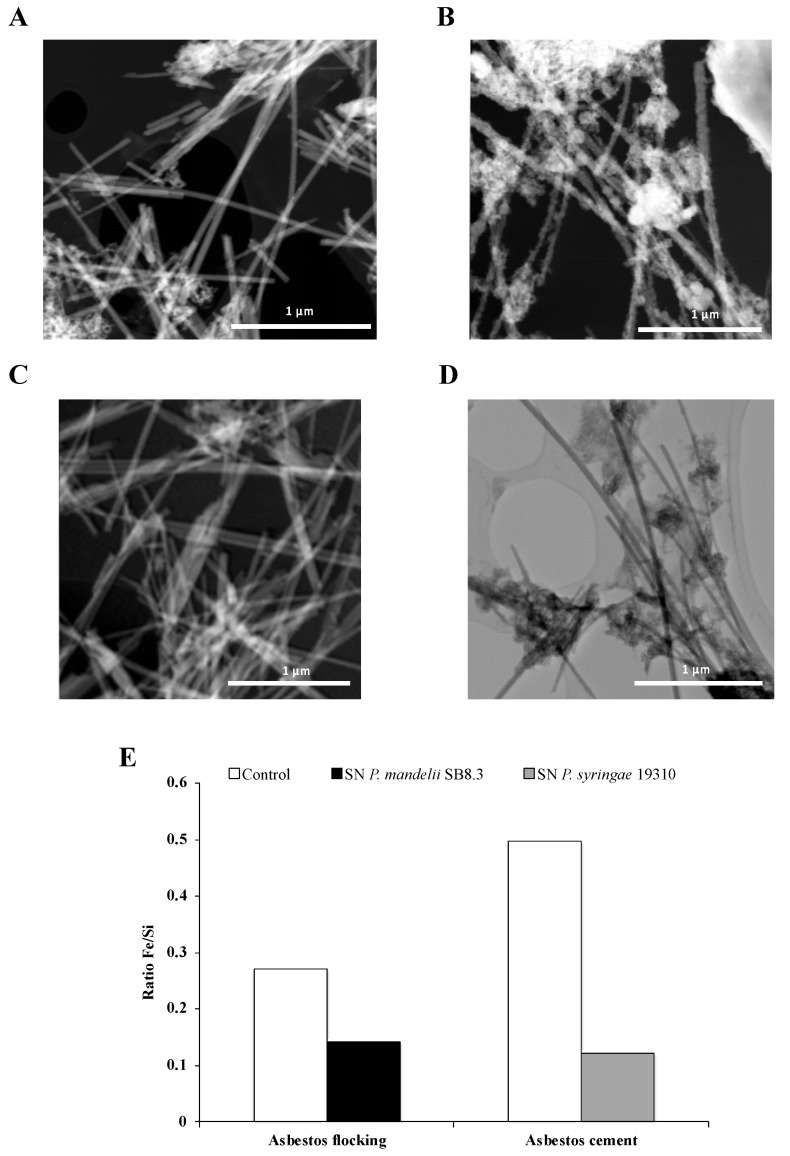
STEM images of chrysotile fibers from asbestos flocking and asbestos cement after respectively 42 and 20 days of total incubation in the presence of succinate (**A**) or casamino acids (**B**) medium as a control, *Pseudomonas mandelii* (**C**) or *Pseudomonas syringae* (**D**) pyoverdine-containing supernatants (SN). Atomic ratios of Fe/Si of total area (**E**). Adapted from David et al. [72,80].

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
