# Peer review of "A Review of Asbestos Bioweathering by Siderophore-Producing Pseudomonas: A Potential Strategy of Bioremediation"

_microorganisms, 2020, doi:10.3390/microorganisms8121870_

Round 1

Reviewer 1 Report

The manuscript is prepared very well.  Taking into account the current problems with asbestos removal  the results presented in the manuscript are very interesting and promising.

I did not find any drawbacks in the manuscript. It is a comprehensive and valuable work.

Reviewer 2 Report

This review paper, focused heavily on the work of the author himself, is an update on the ability of Pseudomonas as a candidate for detoxification of asbestos containing wastes. The review begins well but becomes harder to follow from section 4 where the author begins to introduce aspects of their work (particularly graphs) without a description of the experiment that they came from.

General comment: six of the eight figures were adapted from the author’s previous work. I don’t know enough to comment if the work and contribution of other groups work is sufficiently represented but I do feel like the author is perhaps over familiar with their own data. The review would be clearer if the data presented in the graphs was preceded by a brief description of the experiment they came from and a key to explain the abbreviations used without having to scour the text. I can see that most of the information is there but it breaks up flow of the paper to have to look for it.

Abstract

Line 11: “are highly toxic due to the presence of the transition metal iron.”

I suggest “are highly toxic due primarily to the presence of high concentrations of the transition metal iron”

Iron in small quantities is not toxic. I suggest changing the sentence to clarify we are talking about the effects of high concentrations of iron in specific forms.

Line 22: Suggest “Asbestos is an industrial term referring to six naturally occurring fibrous silicate minerals from the Serpentine and amphibole groups.”

Line 23: insert ‘the’  “from the serpentine group”

Line 31: suggest “the presence of iron in fibres, of up to 30 wt%, is typically correlated… “

State the max percentage to show the readers we are talking about a lot of iron (I got this figure from doi: 10.1038/srep01123 … you probably have other sources)

Line 39: “development of the eco-friendly management”.  Question: are there any non eco friendly management strategies (e.g. chemical dissolution) or is burial the only current solution?

Line 74: state the novelty. Why is this strain being studied? Is it a potential candidate for engineered/stimulated bio reduction? Has it not been studied in relation to asbestos before (you state it has been used with other minerals).

Line 77: This sentence is a bit odd. The definition of a bedrock “the solid rock underlying unconsolidated surface materials (such as soil) 2a : lowest point.”  

I suggest: Asbestiform minerals are found in the bedrock at various locations around the world.

Figure 1: I like this figure, it supports the text well.

Line 107: You mention acid treatment here. It would be worth discussing further up in the introduction where you talk about environmentally friendly treatments.

Line 125: it would be good at this point to define the biological definition of affinity and its units as you go on to used this concept again further down.

Figure 3 caption ‘ground raw chrysotile’ (rather than grinded)

Figures Figure 3, Figure 4 and Figure 5. These figures are hard to understand as too many acronyms are introduced that relate to David et al.’s work. Some you can only find defined later in the text so you have to read ahead to understand. Remember that the reader may not be familiar with David et al and therefore these graphs are hard to decipher.

Example: in figure 3 the abbreviation SN isn’t defined until figure 6. 

I suggest swapping abbreviations for full descriptions where possible or adding a key to the graphs, or describing the experiment more fully in the text BEFORE showing the figure.

Reviewer 3 Report

The article may be accepted for the publication as it presents the information on the relevant issue (novel application of microbial biotechnology to harmful waste treatment). Nevertheless, the article requires major revision as additional information on the issue should be included in the manuscript. Authors did not provide any information on currently used technologies for ACW utilization and their advantages/disadvantages (rate of ACW destruction using different methods, energy consumption, formation of secondary harmful products). This issue should be discussed in the article for understanding potential role of biotechnological method proposed.

Round 2

Reviewer 3 Report

The authors have improved the manuscript according to the reviewer recommendations. Therefore, the article may be accepted.